# Adult Acquired Flatfoot Deformity: A Narrative Review about Imaging Findings

**DOI:** 10.3390/diagnostics13020225

**Published:** 2023-01-07

**Authors:** Chiara Polichetti, Maria Ilaria Borruto, Francesco Lauriero, Silvio Caravelli, Massimiliano Mosca, Giulio Maccauro, Tommaso Greco, Carlo Perisano

**Affiliations:** 1Orthopedics and Trauma Surgery Unit, Department of Ageing, Neurosciences, Head-Neck and Orthopedics Sciences, Università Cattolica del Sacro Cuore, Fondazione Policlinico Universitario A. Gemelli IRCCS, 00168 Rome, Italy; 2Department of Radiological and Hematological Science, Section of Radiology, Università Cattolica del Sacro Cuore, Fondazione Policlinico Universitario A. Gemelli IRCCS, 00168 Rome, Italy; 3IRCCS Istituto Ortopedico Rizzoli—U.O.C. II Clinic of Orthopaedics and Traumatology, 40136 Bologna, Italy

**Keywords:** foot and ankle, adult acquired flatfoot deformity, progressive collapsing foot deformity, imaging, weight bearing CT, posterior tibial tendon

## Abstract

Adult acquired flatfoot deformity (AAFD) is a disorder caused by repetitive overloading, which leads to progressive posterior tibialis tendon (PTT) insufficiency. It mainly affects middle-aged women and occurs with foot pain, malalignment, and loss of function. After clinical examination, imaging plays a key role in the diagnosis and management of this pathology. Imaging allows confirmation of the diagnosis, monitoring of the disorder, outcome assessment and complication identification. Weight-bearing radiography of the foot and ankle are gold standard for the diagnosis of AAFD. Magnetic Resonance Imaging (MRI) is not routinely needed for the diagnosis; however, it can be used to evaluate the spring ligament and the degree of PTT damage which can help to guide surgical plans and management in patients with severe deformity. Ultrasonography (US) can be considered another helpful tool to evaluate the condition of the PTT and other soft-tissue structures. Computed Tomography (CT) provides enhanced, detailed visualization of the hindfoot, and it is useful both in the evaluation of bone abnormalities and in the accurate evaluation of measurements useful for diagnosis and post-surgical follow-up. Other state-of-the-art imaging examinations, like multiplanar weight-bearing imaging, are emerging as techniques for diagnosis and preoperative planning but are not yet standardized and their scope of application is not yet well defined. The aim of this review, performed through Pubmed and Web of Science databases, was to analyze the literature relating to the role of imaging in the diagnosis and treatment of AAFD.

## 1. Introduction

Acquired adult flatfoot deformity (AAFD), more recently defined as progressive collapsing foot deformity (PCFD) [1,2,3,4], is a deformity characterized by a lack of propulsive gait and a partial or complete flattening of the medial arch of the foot in weight bearing that develops after skeletal maturity [1,2]. AAFD affects mostly middle-aged and elderly women, with high body mass index (BMI), resulting in foot pain, malalignment, loss of function and impact on quality of life [5,6,7,8]. A survey in the UK of women over 40 years old estimated the prevalence to be over 3% [1]. Several factors have been proposed in the etiology of adult-acquired flatfoot deformity including arthritic, neuromuscular, and traumatic conditions [9]; however, posterior tibial tendon (PTT) dysfunction, principally due to micromechanical trauma of repetitive loading [10], remains the most common etiology [11]. Intrinsic and extrinsic factors are implicated in the development of AAFD. Extrinsic factors such as obesity, foot shape, acute traumatic injury, and equinus contracture of the gastrocnemius-soleus complex can increase the force experienced by the PTT, exposing it to more mechanical trauma. Pre-existing deformities, such as asymptomatic flexible pes planus, accessory navicular, or valgus orientation of the subtalar joint, can also make the foot more susceptible to AAFD. Intrinsic factors, inflammatory disorders, hypertension, and diabetes mellitus, can predispose the tendon to depletion and degeneration [10]. Nevertheless, the involvement of other soft tissue of the hindfoot and medial longitudinal arch, including the spring, deltoid, and interosseous talocalcaneal ligaments, has also been shown to play a role in the development of AAFD [12]. AAFD is considered pathological only when symptomatic. Pain is localized in different parts of the foot depending on the triggering causes. Generally, it is located in the medial part of the hindfoot, along the posterior tibial tendon, sometimes associated with effusion into the tendon sheath, but it may be plantar and deep in spring ligament lesion or also lateral due to fibula-calcaneal or calcaneocuboid impingement [6]. Diagnosis requires clinical history assessment, physical examination, and evaluation of imaging findings. Clinical evaluation allows the establishment of the characteristics of the deformity, namely whether it is still reducible and whether it maintains a range of motion (ROM). 

The aim of this review is to summarize the features and parameters that can be assessed with imaging techniques, both traditional and more innovative, that can help the clinician in the diagnosis and management of the patient with AAFD. 

## 2. Imaging Techniques

### 2.1. Radiography

Radiography is the first-level imaging exam which should be performed in antero-posterior (AP) and lateral weight-bearing (WB) views of both feet (also in case of a unilateral deformity, to compare measurements with normal side), as well as standing AP views of both ankles and the view of Saltzman [5,13]. The X-ray detector and tube should be 35–40 inches apart in both the AP and lateral views of the foot and ankle; furthermore, the beam must be angled at 10° when the AP view of the foot is performed [14]. In Saltzman view, also called hindfoot alignment view, subjects stood on a radiolucent platform with equal weight on both feet. The X-ray tube is oriented 20° from the horizontal, so that it is perpendicular to the plane of the film. The beam is centered at the level of the ankle, and the field of exposure included from midshaft of the tibia to below the calcaneus. Exposure is 62 KV, 6 Mas with a 400 speed, non-grid system using an 11 × 14 in film [13,15]. 

Numerous radiographic measurements have been proposed for assessing the structural changes associated with AAFD. 

In the AP view [7,10,11,16] the following can be measured and evaluated:*Talar–first metatarsal angle*, between the lines drawn along the long axis of the talus and the first metatarsal (normal 0°, flatfoot: mild >4°, moderate >15°, severe >30°) (Figure 1);*Talonavicular coverage angle*, between the line that joins the medial and lateral articular margins of the talus, and the line that joins the medial and lateral articular margins of the navicular, it represents forefoot abduction (normal, <7°; flatfoot, >7°) (Figure 2);*Talonavicular uncoverage percentage*, the percentage of the talus that is not in contact with the navicular medially, useful to evaluate forefoot abduction (normal, 10% to 30%; flatfoot >30%) (Figure 3);*Talar incongruency angle* (normal, 5° = –26°; flatfoot >26°), is formed by the intersection between a line from the most lateral point of the articular surfaces of the talus and the navicular, and a line from the lateral aspect of the talar neck (in its most narrow segment) to the lateral point of the talar articular surface (Figure 4).

In lateral view [7,10,11]:*Talar–first metatarsal angle* (Meary’s angle), the normal value is 0 ± 10 degrees and is increased in flatfoot deformity (often >20°, apex directed plantarly) (Figure 5);*Calcaneal pitch,* the angle between the line parallel to the ground and the line along the inferior inclination axis of the calcaneus (normal, 20–30°; flatfoot, <20°) (Figure 6);*Talocalcaneal angle* is formed by the long axis of the rearfoot and the midtalar line. This angle is increased in pronated feet on both the AP and lateral views (normal <45°, flatfoot >45°) (Figure 7);*Calcaneal-fifth metatarsal angle*, defined as the angle formed between the tangent to the inferior aspect of the calcaneus and a line drawn along the inferior aspect of the base and head of the fifth metatarsal (normal <170°, flatfoot >170°) (Figure 8).

In Saltzman view (Figure 9):*Hindfoot moment arm*, measured by the shortest distance between the midtibial axis and the most inferior portion of the calcaneus gus (normal, −3 mm to +10 [varus]; flatfoot, >+10 mm [valgus]);*Hindfoot alignment angle*, formed by the intersection of the longitudinal axis of the tibial shaft and the axis of the calcaneal tuberosity (normal, 5.6 ± 5.4°; flatfoot, 22.5 ± 4.9°) [17].

Considering their widespread use in clinical practice, many studies have tried to quantify the reliability of the angular measurements using this conventional imaging. Sensiba et al. [18] investigated the intra- and inter-observer reliability using three digital radiographs consisting of AP, lateral, and hindfoot alignment, varying levels of observer experience. Intra-observer reliability increased with observer experience. Nevertheless, traditional images are not able to show soft tissue involvement or bone oedema. Other radiographic techniques are often necessary to establish the etiology of the disease and the preoperative planning [12].

The radiographic study was also useful for assessing post-operative outcomes, although the different measurements showed different reliability depending on the corrective surgical technique used [19]. 

In particular, in a medializing calcaneal osteotomy, some authors found significant improvements in radiographic measures of midfoot abduction and medial longitudinal arch [19], whereas others found no change in Meary’s angle and calcaneal pitch postoperatively [20,21]. In the case of lateral column lengthening (LCL), Sangeorzan et al. found an improvement in the AP talar–first metatarsal angle and in the talonavicular coverage angle [22]; other authors found that the LCL was the only significant contributor to the change in the lateral incongruency angle [22,23]. With regard to the opening medial cuneiform osteotomy (Cotton osteotomy), the best indicator of correction proved to be lateral Meary’s angle in the immediate post-operative and during the early follow-up; however, there was statistically significant loss of correction between intermediate and final radiographs [24]. Instead, regarding triple arthrodesis, authors showed an improvement in Meary’s angle and in calcaneal pitch, kept at 24 months follow-up [24,25,26].

### 2.2. Magnetic Resonance Imaging (MRI)

MRI is useful to analyze tendon involvement, soft tissue, and bone structures, allowing the study of the physiopathology of AAFD [7,27,28]. A common incidental finding of MRI is the lateral ankle ligament injury which is secondary to the biomechanical changes within the foot and increases stress placed on the soft tissues surrounding both ankle and subtalar joints, causing pain [29].

MRI protocols should include standard sequences for studying the ankle such as sagittal T1-weighted spin-echo, sagittal fast spin-echo short tau inversion recovery (STIR), axial proton-density weighted, axial T2-weighted fast spin echo fat-saturated, and coronal proton-density weighted fat-saturated. High field strength 1.5T or 3T scanners and dedicated extremity coils provide adequate signal-to-noise ratio and the spatial resolution required for proper imaging of the complex anatomy of the ankle [14]. 

#### 2.2.1. Posterior Tibial Tendon

MRI is the preferred modality for assessment of the PTT (Figure 10a) and it has a good accuracy in showing tendon abnormalities, with a sensitivity of up to 95% and a specificity of 100% in the detection of the rupture of the PTT [30]. The PTT can be affected by degenerative tendinosis, tenosynovitis, and a partial or complete tear. PTT tendinosis can be diagnosed with increased signal intensity in the tendon with preservation of the normal shape; tenosynovitis (isolated) is documented if there is a high fluid signal intensity surrounding 50% or more of the PTT circumference with normal size, shape, and intrinsic low signal. A PTT tear is identified with increased T2 and T1 signal intensity and enlargement of the tendon, defects in the tendon, a linear fluid signal in the tendon (longitudinal split tear) or a lack of continuous tendon fibres (complete rupture) [14].

#### 2.2.2. Spring Ligament 

The spring ligament (Figure 10b), also known as the plantar calcaneo-navicular ligament, is considered the primary static stabilizer of the medial arch and is second in importance only to the PTT [7]. The superomedial component of the spring ligament complex is the largest and most important part and constitutes a structure that supports the talar head and talonavicular joint and separates the PTT from the talus bone. When it is injured or released and cyclically loaded, it leads to a weakening of other structures, resulting in a plano-valgus deformity [31]. MRI is useful to study the injuries of this ligament and their severity. Diagnostic MRI findings include abnormally high signal intensity on T2-weighted or proton density images, thickening (>5–6 mm), thinning (<2 mm), waviness, and discontinuity [32]. Ormsby et al. showed that in the progression of injury related to AAFD, the dorsal talonavicular component of the tibio-navicular ligament can be involved and it can be well shown by new MRI sequences, which consists of an axial oblique proton dense fat suppressed sequence with 3-mm (0.3 mm gap) slices at 30° to the axial plane along the tibionavicular portion of the deltoid complex [33].

Postoperatively, MRI may be employed for patients who have persistent pain after surgical correction for AAFD. It allows observation of osteotomy healing, the integrity of soft-tissue reconstructions, insufficiency fractures, and infection. In soft-tissue reconstructions, through MRI, integrity and remodelling of the graft can be shown, with the transposed tendon becoming more well-defined and hypointense over time [34].

### 2.3. Ultrasound (US)

US is an inexpensive but operator-dependent investigation, assessing PTT, spring ligament status (Figure 11) and foot anatomy-functional changes [35], with similar accuracy to MRI [11]. Dynamic Mode can be useful in patients with suspected friction syndrome [7], at a thickened retinaculum and tendon instability related to flexor retinaculum disruption, which fosters anterior tendon subluxation.

### 2.4. Computed Tomography (CT)

CT provides advanced and detailed visualization of the hindfoot and it is useful in the evaluation of bone abnormalities, such as arthritis, tarsal coalitions, and fracture malunions [36]. However, with the traditional CT, images can be obtained only offload. In patients with AAFD, hindfoot instability and alignment have been better observed during WB [37]. Recent developments in CT scan design have contributed to the advent of Cone-beam computed tomography (CBCT) which allows imaging of lower extremities in a normal upright WB. CBCT uses a large-area detector and a pyramid-shaped X-ray beam, obtaining fully volumetric data from multiple projections acquired in a single rotation about the patient without moving the patient through the scanner [38].

Through CT scans, numerous measurements, which can also be calculated with X-ray, can be obtained with a higher level of accuracy [12].

In the axial plane (Figure 12):
the *talar-first metatarsal angle;*the *talonavicular coverage angle.*

In the coronal plane, nine parameters can be evaluated: the *forefoot arch angle*;the *navicular-to-skin distance*;the *navicular-to-floor distance*;the *medial cuneiform-to-skin distance*;the *medial cuneiform-to-floor distance*;the *calcaneofibular distance*;the subtalar horizontal angle, which is the angle between the posterior facet of the talus and the floor measured at 25% (posterior aspect), at 50% (midpoint), and at 75% (anterior aspect) of the posterior subtalar joint length.

In the sagittal plane, eight parameters can be assessed:the *talus-first metatarsal angle*;the *navicular-to-skin distance*;the *navicular-to-floor distance*;the *cuboid-to-skin distance*;the *cuboid-to-floor distance*;the *medial cuneiform-to-skin distance*;the *medial cuneiform-to-floor distance*;the *calcaneal inclination angle*.

Several studies [12,38,39,40,41] reported excellent image quality with sufficient contrast resolution to visualize soft tissue and bone. The recent advent of 3D reconstruction provides more precise morphologic analysis and better evaluation of the subtalar joint surface of the calcaneus by talar subtraction and contributes to any surgical treatment planning [6]. 

#### Weightbearing CT (WBCT)

WBCT showed a low radiation exposure (10% to 66%) compared to conventional multidetector CT scanners, therefore it may be advantageous for those patients who require frequent imaging studies. Patients are scanned in a physiological standing WB position, with their feet at shoulder width and distributing their body weight equally between both lower extremities [41]. 

WBCT provides additional parameters used to quantify flatfoot deformity and may identify underlying anatomic variants in the subtalar joint that predispose to peritalar subluxation and flatfoot. It may help identify specific locations of deformity (e.g., medial arch collapse at the talonavicular or naviculocuneiform joint) and localize impingement, arthritis, or severe calcaneal subluxation, which may need to be addressed with hindfoot (i.e., subtalar) fusion [19]. WBCT has also proven useful in the evaluation of stiff flatfoot because it allows exploration of talocalcaneal or calcaneonavicular synostosis coalition in young patients; subtalar, tibiotalar, talonavicular, or Lisfranc joint osteoarthritis in older patients; and in the measurement of foot and ankle offset (FAO), a multiplanar measurement that assesses the relationship between the center of the ankle joint and the weight-bearing tripod of the foot, consisting of the first and fifth metatarsal head and calcaneus [6,42]. FAO is calculated using software where the operator inserts a value obtained through landmark coordinates on multiplanar reconstruction images, where a normal value is 2.3% ± 2.9%, and it is used to evaluate operative treatment deformity correction. [16]. WBCT has recently become an important diagnostic imaging tool to evaluate not only the severity of PCFD, but also the widening and instability of distal tibiofibular syndesmotic (DTFS) injuries [1]. The impingement between the talus and/or calcaneus with the distal fibula, in the setting of a chronic hindfoot valgus deformity and progressive peritalar subluxation in patients with AAFD, would potentially lead to high stresses in the DTFS ligaments and joints, causing widening of the syndesmotic space and possible syndesmotic instability. 

From the analysis of the available literature, CT does not have a role in the assessment of postoperative outcomes, although it finds wide use in the evaluation of possible postoperative complications [43]. Day et al. used WBCT to evaluate AAFD correction by assessing FAO, which was significantly influenced and improved, among the multiple operative procedures performed, by reconstruction of the spring ligament only [42].

## 3. Discussion

Staging AAFD, assessing both soft tissue and bone changes, is necessary to plan suitable surgery treatment [43]. Although new imaging techniques are making their way into AAFD diagnostics, radiography remains the most widely used and easiest imaging technique. Consequently, the available literature provides much information on traditional X-ray, CT, and on modern techniques such as CBCT and WBCT, although some of these are not yet standardized. Many articles have analyzed the potential and limits of CT, while there is still a paucity of studies on MRI. Furthermore, in the literature, the role of MRI images is mostly to analyze pathogenesis rather than diagnostic or prognostic [7,29,33]. New MRI sequences which evaluate ligament involvement and bone oedema are available.

Radiographic measurements are used principally to evaluate longitudinal arch flattening, hindfoot valgus, and forefoot abduction. The most used measurements for the longitudinal arch are the Meary angle, the calcaneal pitch, and the calcaneal–fifth metatarsal angle. The most common metrics for hindfoot valgus and forefoot abduction are the talocalcaneal angle, the talus–first metatarsal axis, and the talonavicular angle [7,10,11,24]. Nevertheless, numerous other parameters were devised for assessing the angular changes associated with AFFD. Except for medial cuneiform-first metatarsal angle, these measurements have shown great inter observer reliability [37]. Indeed, the 2-dimension nature of plain radiographs and the need for these of calibration, limits their accuracy and optimal evaluation of AAFD. Standard imaging may miss critical deformity’s information that may translate into diagnostic and treatment opportunities. Traditional imaging in AAFD has problems with reproducibility and accuracy mostly because of the difficulties in studying a 3-dimensional deformity using 2-dimensional images. Nowadays, newer techniques which allow 3-D imaging, such as WB and simulated WB CT, provide further information about the deformity.

WBCT scans have been used to quantify the severity of deformities in AAFD with good intra-observer and inter-observer reliability, and the results favored WB images over non-weight-bearing (NWB) images. The literature agrees on the greater reliability of WBTC compared with all other techniques, particularly on the evaluation of measurements especially useful for pathology staging and preoperative planning. According to several studies led by De Caesar Netto et al [3,12,44], this may be attributable to the real floor line that was definable on the WB images but not on the NWB images. In addition, measurements of distances were more reliable than those of angles. In fact, the addition of a second line to build an angle increased intra- and inter-observer variability [37]. The measurements performed in the axial plane demonstrated lower reliability than those in the other planes with the lowest inter-observer reliability for the talus-first metatarsal angle in both the sagittal and the axial plane. It was found that the least reliable measurement was the talus-first metatarsal angle, or Meary angle, an important index of arch collapse and one of the most helpful to grade AAFD. This limit is due to the difficulty to include the talus and first metatarsal in the same image. Many papers [12,37,45,46] showed that cone-beam CT can demonstrate worsening of AAFD when performed during physiological WB. When compared with NWB images, WB cone-beam CT images show significantly increased deformity as reflected by almost all parameters that were evaluated. On the axial view, the talus-first metatarsal angle and talonavicular coverage angle increased by 57% and 43%, respectively, reflecting increased abduction of the hindfoot. In the coronal plane, WB images showed a 78% decrease in the forefoot arch angle; similar results were found for hindfoot valgus deformity parameters. Moreover, some of the differences between the measurements on the NWB and WB images, even when they were statistically significant, might not be clinically important and may have been caused by measurement error [12,37]. Further studies are needed to evaluate the use of WB 3D cone-beam CT in diagnostic and therapeutic practice and to assess correction of the deformity after surgery, although the limited availability of WBCT in various hospitals limit its use, despite its many benefits and advantages.

MRI is not routinely needed for the diagnosis of adult-acquired flatfoot deformity. For some investigators, MRI is the method of choice in the evaluation of the pathology of the PTT, although its exact role in the treatment plan remains controversial and not clearly defined. Other investigators strongly believe that MRI has a distinctive role, particularly in cases in which the diagnosis is unclear [36]. Furthermore, the use of CT and MRI was compared in the evaluation of PTT degeneration and correlated with surgical findings. The percentage of tears that were diagnosed was greater in the group who underwent MRI (73%) compared with CT (59%) [30].

## 4. Conclusions

X-ray techniques help in analyzing and quantifying the deformity and are still the gold standard because of their cost and simplicity of execution; however, they lack reproducibility and accuracy. Further techniques, such as CT, CBCT, WBCT, and MRI, are useful to understand the deformity and the symptoms. However, currently, the use of these new methods is limited, especially for WBCT, which is not available in many centers although it is gaining ground and more widespread use. In the future, it may improve surgical planning and assess outcomes.

## Figures and Tables

**Figure 1 diagnostics-13-00225-f001:**
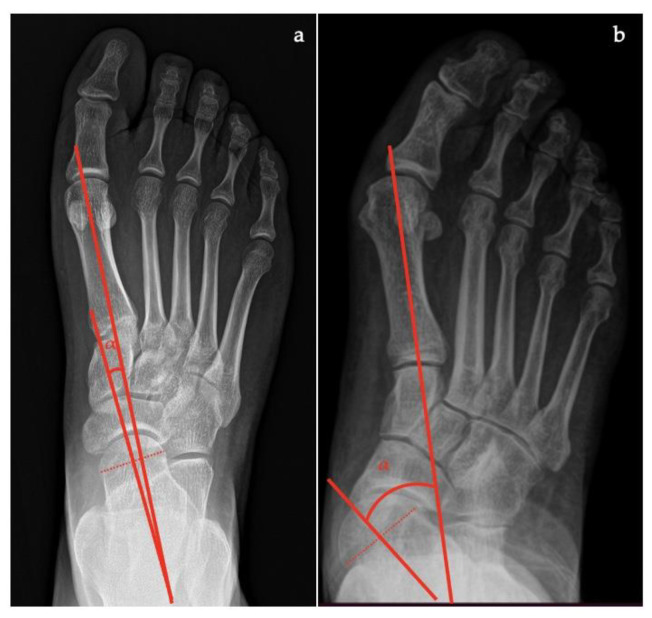
Taler-first metatarsal angle in the anteroposterior view (between the long axis of the talus and the long axis of the first metatarsal). (**a**) Normal foot, 3°; (**b**) pathological flatfoot, 30°.

**Figure 2 diagnostics-13-00225-f002:**
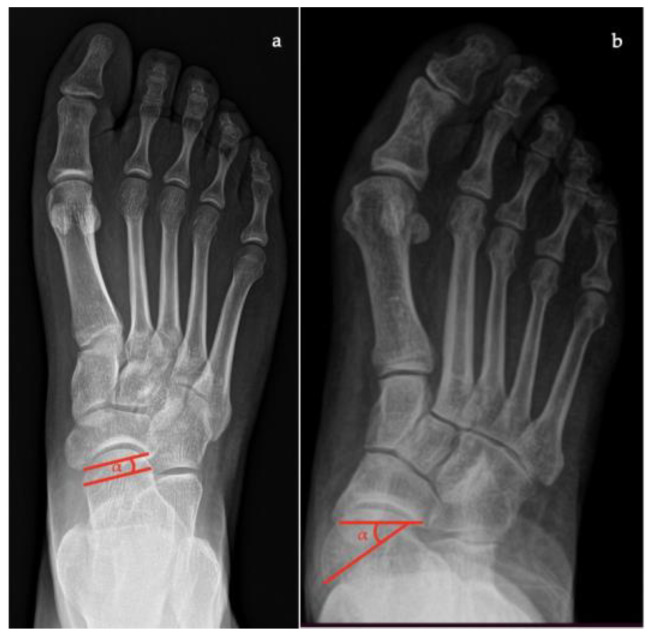
Talonavicular coverage angle (between the line that joins the medial and lateral articular margins of the talus, and the line between that joins the medial and lateral articular margins of the navicular). (**a**) Normal foot, 2°; (**b**) pathological flatfoot, 35°.

**Figure 3 diagnostics-13-00225-f003:**
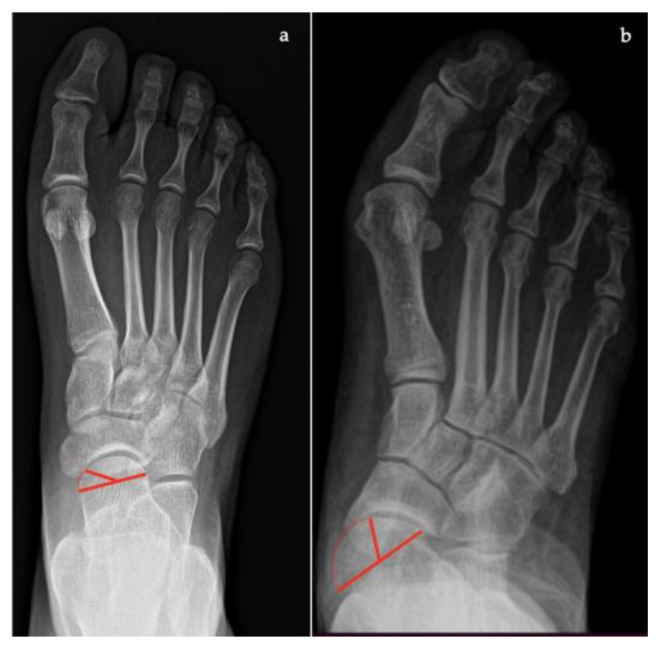
Talonavicular uncoverage percentage (percentage of the talus that is not in contact with the navicular medially). (**a**) Normal foot, <30%; (**b**) pathological flatfoot, >30%.

**Figure 4 diagnostics-13-00225-f004:**
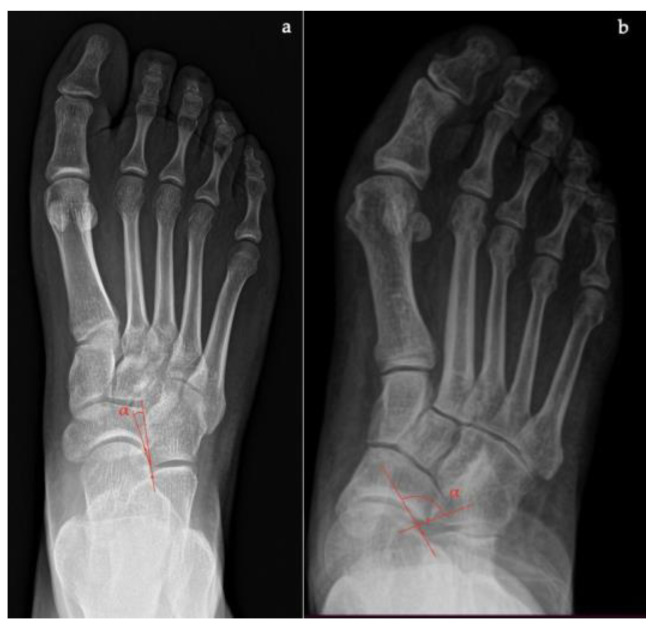
Talar incongruency angle (formed by the intersection between a line from the most lateral point of the articular surfaces of the talus and the navicular, and a line from the lateral aspect of the talar neck (in its most narrow segment) to the lateral point of the talar articular surface). (**a**) Normal foot, 6°; (**b**) pathological flatfoot, >30°.

**Figure 5 diagnostics-13-00225-f005:**
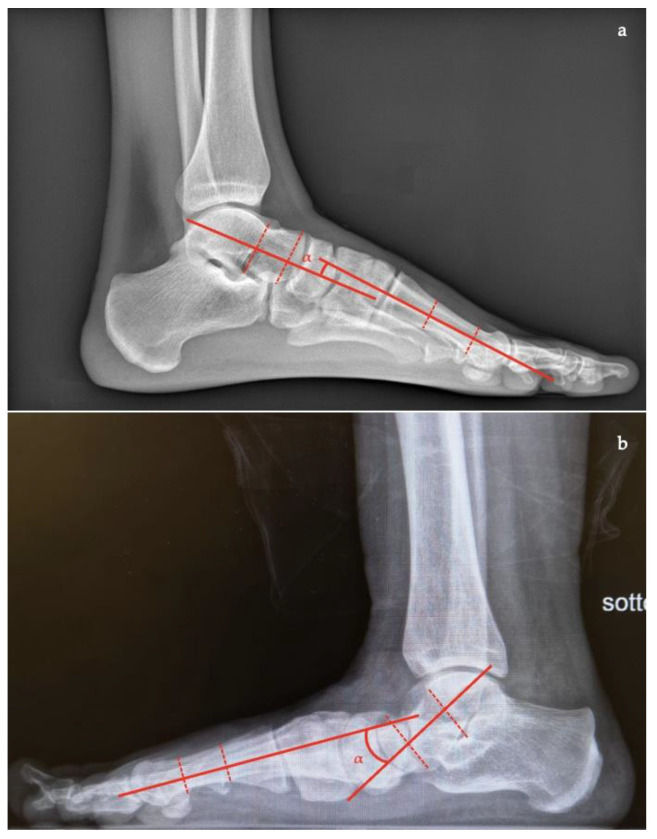
Talar-first metatarsal angle in the lateral view (Meary’s angle, the angle between the long axis of the talus and the long axis of the first metatarsal). (**a**) Normal foot, 4°; (**b**) pathological flatfoot, 30°.

**Figure 6 diagnostics-13-00225-f006:**
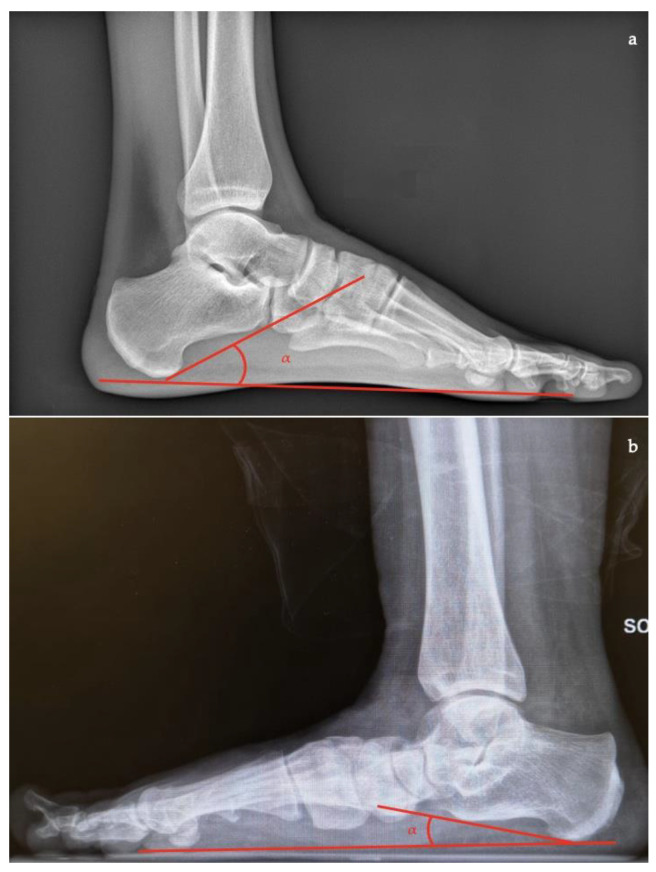
Calcaneal pitch (formed by the line parallel to the ground and the line along the inferior inclination axis of the calcaneus) in the lateral X-ray view. (**a**) Normal foot, 30°; (**b**) pathological flatfoot, 10°.

**Figure 7 diagnostics-13-00225-f007:**
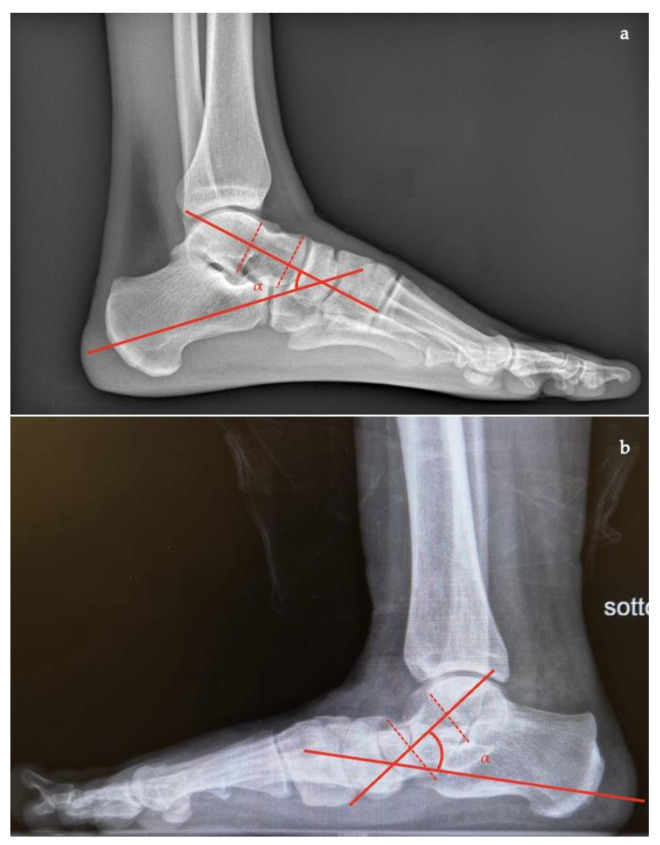
Talocalcaneal angle (formed by the long axis of the rearfoot and the midtalar line) in lateral X-ray view. (**a**) Normal foot, 45°; (**b**) pathological flatfoot, 52°.

**Figure 8 diagnostics-13-00225-f008:**
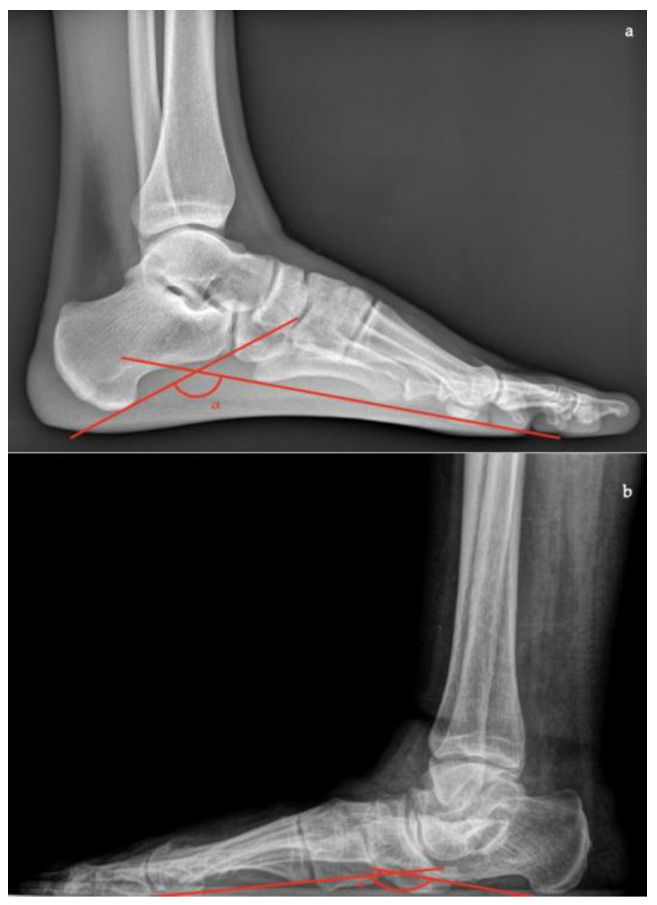
Calcaneal-fifth metatarsal angle (defined as the angle formed between the tangent to the inferior aspect of the calcaneus and a line drawn along the inferior aspect of the base and head of the fifth metatarsal) in the lateral X-ray view. (**a**) Normal foot, 145°; (**b**) pathological flatfoot, 175°.

**Figure 9 diagnostics-13-00225-f009:**
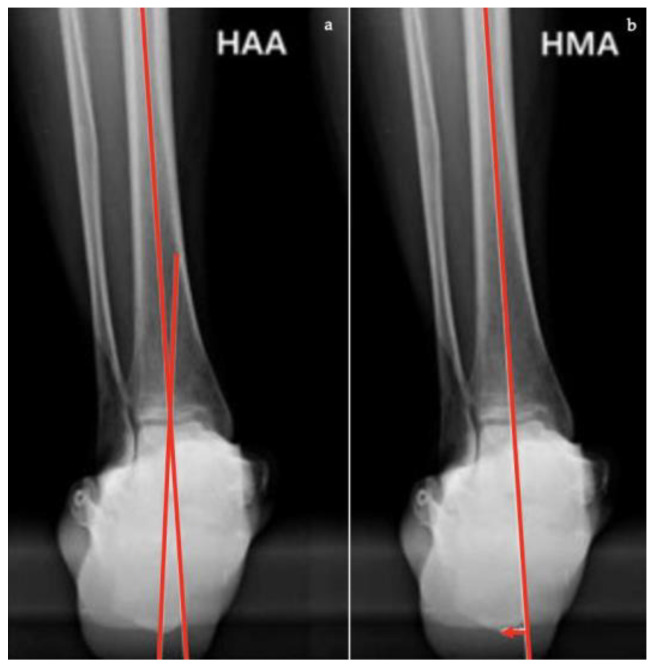
Saltzman view. (**a**) Hindfoot Alignment Angle (HAA) (formed by the intersection of the longitudinal axis of the tibial shaft and the axis of the calcaneal tuberosity). (**b**) Hindfoot Moment Arm (measured by the shortest distance between the midtibial axis and the most inferior portion of the calcaneus gus, represented by the red arrow).

**Figure 10 diagnostics-13-00225-f010:**
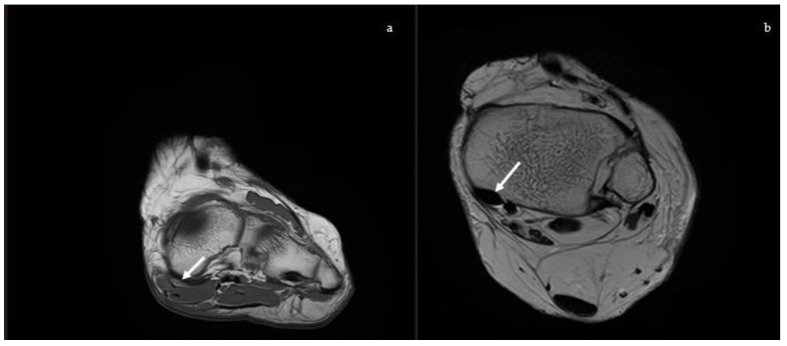
Magnetic Resonance Imaging. (**a**) Coronal T1: soft tissue involvement of the posterior tibial tendon, (**b**) Axial T2: superomedial fibers of the spring ligament (white arrow).

**Figure 11 diagnostics-13-00225-f011:**
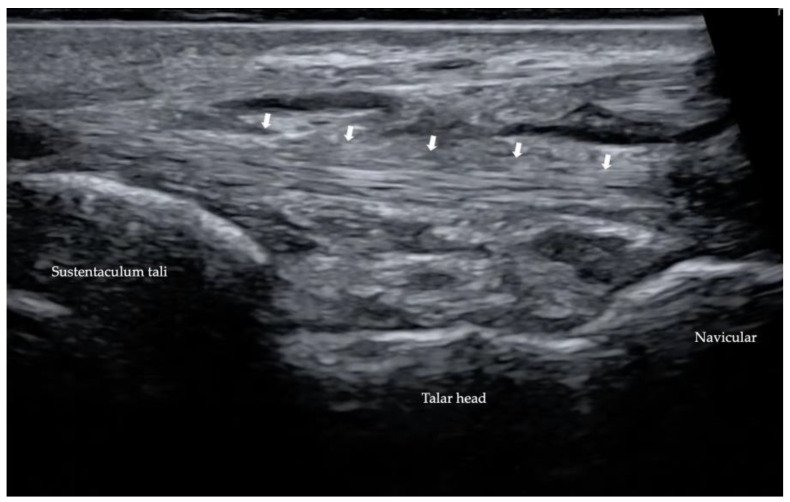
Ultrasonography of spring ligament (marked by white arrows).

**Figure 12 diagnostics-13-00225-f012:**
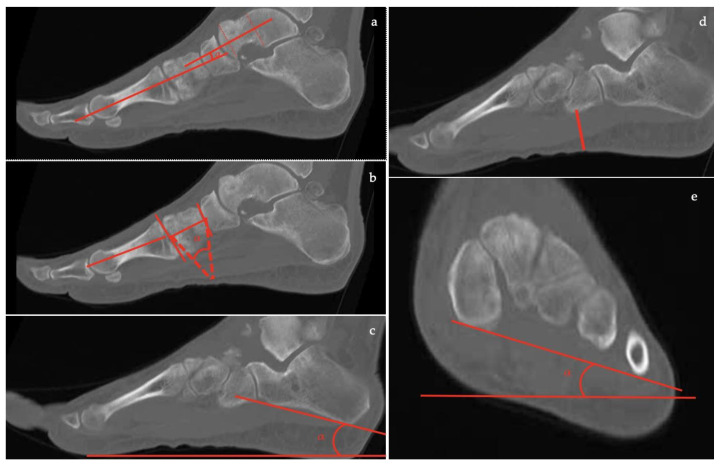
Computed Tomography (CT) imaging: (**a**) Talar–first metatarsal angle (Meary’s angle), (**b**) Medial Cuneiform-First metatarsal angle, (**c**) Calcaneal inclination angle, (**d**) Cuboid to floor distance, (**e**) Forefoot arch angle.

## Data Availability

Not applicable.

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
