# Peer review of "Adult Acquired Flatfoot Deformity: A Narrative Review about Imaging Findings"

_diagnostics, 2023, doi:10.3390/diagnostics13020225_

Round 1

Reviewer 1 Report

L122-133.- it is inappropriate to speak of surgery technics here. Only you cites some technics. I think that it is unnecessary  

It will be good an image of the spring ligament in the section ecography 

In the discussion, when it is spoken of the measurement more used, you need to put more than one references

Author Response

L122-133.- it is inappropriate to speak of surgery technics here. Only you cites some technics. I think that it is unnecessary  

-Thanks for the suggestion, which we appreciate. As correctly pointed out by the reviewer, the surgical treatment of AAFD is beyond the scope of our review, so we have not covered the technical part of the various surgical approaches in detail, but we believe that a little indication on the use of imaging examinations in the post-operative period may be useful. We hope we have clarified the reasons for our choice.

It will be good an image of the spring ligament in the section ecography 

-Thank you for yout suggestion; an US image of the spring ligament has been added (Figure 11).

In the discussion, when it is spoken of the measurement more used, you need to put more than one references

- Thank you for the tip. We have added more references in the discussion about measurement more used.

Reviewer 2 Report

I have been invited to review the manuscript titled “Adult acquired flatfoot deformity: imaging findings”. This is a topic of interest in current research and has potential to spread interesting results. However, there are issues that must be addressed in order to improve the quality of the manuscript.

Title

The title should be amended slightly to ensure that the reader understands the type of research  immediately.

Abstract

The databases used when the search was carried out must be named.

Introduction

Flatfoot deformity  is one of the most frequent musculoskeletal injuries, but to include the position it holds would be of value and the actual impact in the quality of life, see the reearch of foot arch height and quality of life in adults https://pubmed.ncbi.nlm.nih.gov/30041462/ 

 The background shown about different possibilities when 

 literature relating the role of imaging in diagnosis and treatment is too short, thus it would be necessary to increase knowledge in this regard. Thus, I suggest in this section should be improved, with more details about the research of Calvo Lobo  et al ultrasound evaluation of intrinsic plantar muscles and fascia in hallux valgus: A case-control study https://pubmed.ncbi.nlm.nih.gov/27828846/ and Geometry of the Proximal Phalanx of Hallux and First Metatarsal Bone to Predict Hallux Abducto Valgus: A Radiological Study https://pubmed.ncbi.nlm.nih.gov/27861517/

Moreover, to highlight why this systematic review is needed should be indicated.

Methods and Results. 

Add these sections and respond the following questions:

Was the study registrered at PROSPERO? 

Design of the study: This must be properly named

The PICO question should be clearly indicated.

Why did authors choose only systematic reviews or meta analysis when eligibility criteria?

What is the search strategy and Which several databases were used?

Discussion

Include this section the principal strengths and weaknesses in relation to other studies, discussing important differences in results; the meaning of the study: possible explanations and implications and unanswered questions and future research

Conclusion

Conclusion should be based on the findings from the study, and not in what gaps were found. Thus, recommendations should be shown is the discussion section and the statements proposed should be better expressed to not misunderstand the real meaning.

Author Response

I have been invited to review the manuscript titled “Adult acquired flatfoot deformity: imaging findings”. This is a topic of interest in current research and has potential to spread interesting results. However, there are issues that must be addressed in order to improve the quality of the manuscript.

-Thanks to the reviewer for the thorough and detailed revision. Corrections  have been made to the manuscript according to his suggestions, highlighting them in yellow.

Title

The title should be amended slightly to ensure that the reader understands the type of research immediately.

- Thank you for your comment. We have changed the title to allow the reader an immediate understanding of the type of study.

New title: Adult acquired flatfoot deformity: a narrative review of imaging findings.

Abstract

The databases used when the search was carried out must be named.

- Thank you for your comments. The review was performed through Pubmed and Web of Science (this information was included to the abstract, line 29).

Introduction

Flatfoot deformity  is one of the most frequent musculoskeletal injuries, but to include the position it holds would be of value and the actual impact in the quality of life, see the reearch of foot arch height and quality of life in adults https://pubmed.ncbi.nlm.nih.gov/30041462/ 

- Thank you for the comment which gives us the opportunity to further explore the impact of pathology on patients' quality of life. The proposed reference, which is very relevant, has been added (reference 8). “AAFD affects mostly middle-aged and elderly women, with high body mass index (BMI), resulting in foot pain, malalignment, loss of function and impact on quality of life [5,7,8].”

 The background shown about different possibilities when 

 literature relating the role of imaging in diagnosis and treatment is too short, thus it would be necessary to increase knowledge in this regard. Thus, I suggest in this section should be improved, with more details about the research of Calvo Lobo  et al ultrasound evaluation of intrinsic plantar muscles and fascia in hallux valgus: A case-control study https://pubmed.ncbi.nlm.nih.gov/27828846/ and Geometry of the Proximal Phalanx of Hallux and First Metatarsal Bone to Predict Hallux Abducto Valgus: A Radiological Study https://pubmed.ncbi.nlm.nih.gov/27861517/

- Thank to the reviewer for the comment and the tips. We have expanded the discussion including the suggested references (16-35). “US is an inexpensive but operator-dependent investigation, assessing PTT, spring ligament status (Figure 11) and foot anatomo-functional changes [35], with a similar ac-curacy to MRI [11].”

Moreover, to highlight why this systematic review is needed should be indicated.

Methods and Results. 

Add these sections and respond the following questions:

Was the study registrered at PROSPERO? 

Design of the study: This must be properly named

The PICO question should be clearly indicated.

Why did authors choose only systematic reviews or meta analysis when eligibility criteria?

What is the search strategy and Which several databases were used?

- We thank the reviewer for questions and suggestions. The purpose of our study is to conduct a narrative review of the literature. Therefore, to make the nature of our paper clearer, we have changed the title. Since it is not a systematic review, we did not include the materials and methods section. We have tried to extract the results and indications for appropriate imaging examinations from studies with the highest possible level of evidence (systematic reviews, where present).

Discussion

Include this section the principal strengths and weaknesses in relation to other studies, discussing important differences in results; the meaning of the study: possible explanations and implications and unanswered questions and future research

- We thank the reviewer for the comment, but if possible, we ask the reviewer to explain his point of view in more detail. In the discussion, centred on the value of the various imaging methods, we analysed the strengths and weaknesses of the various studies and imaging investigations, discussing the indications in the diagnostic approach to pathology, not forgetting possible insights for future research.

Conclusion

Conclusion should be based on the findings from the study, and not in what gaps were found. Thus, recommendations should be shown is the discussion section and the statements proposed should be better expressed to not misunderstand the real meaning.

-Underlining how, from the analysis of the literature, Rx remains the gold standard, while new imaging methods, although promising, are not used in all centres.

Round 2

Reviewer 2 Report

I am happy with the paper as it stands. Congratulations.